# Quantitative live-imaging reveals the dynamics of apical cells during gametophyte development in ferns

Xiao Wu[1,2,3], An Yan[4,5], Xing Liu[2,6], Shaoling Zhang[3,] and Yun Zhou[1,2,] 🆔

[1]Department of Botany and Plant Pathology, Purdue University, West Lafayette, Indiana 47907, USA; [2]Purdue Centre for Plant Biology, Purdue University, West Lafayette, Indiana 47907, USA; [3]Centre of Pear Engineering Technology Research, State Key Laboratory of Crop Genetics and Germplasm Enhancement, Nanjing Agricultural University, Nanjing, China; [4]Division of Biology and Biological Engineering, California Institute of Technology, Pasadena, California 91125, USA; [5]Howard Hughes Medical Institute, California Institute of Technology, Pasadena, California 91125, USA; [6]Department of Biochemistry, Purdue University, West Lafayette, Indiana 47907, USA

## Original Research Article

**Keywords:**
apical cell; cell division; confocal live imaging; ferns; gametophytes; meristems.

**Authors for correspondence:**
S. Zhang, Y. Zhou,
E-mail: slzhang@njau.edu.cn;
zhouyun@purdue.edu

## Abstract

Meristems in land plants share conserved functions but develop highly variable structures. Meristems in seed-free plants, including ferns, usually contain one or a few pyramid-/wedge-shaped apical cells (ACs) as initials, which are lacking in seed plants. It remained unclear how ACs promote cell proliferation in fern gametophytes and whether any persistent AC exists to sustain fern gametophyte development continuously. Here, we uncovered previously undefined ACs maintained even at late developmental stages in fern gametophytes. Through quantitative live-imaging, we determined division patterns and growth dynamics that maintain the persistent AC in *Sphenomeris chinensis*, a representative fern. The AC and its immediate progenies form a conserved cell packet, driving cell proliferation and prothallus expansion. At the apical centre of gametophytes, the AC and its adjacent progenies display small dimensions resulting from active cell division instead of reduced cell expansion. These findings provide insight into diversified meristem development in land plants.

## 1. Introduction

Meristems in land plants share conserved roles in shaping plant body formation. They remain undifferentiated while driving cell proliferation and organ development (Greb & Lohmann, 2016; Heidstra & Sabatini, 2014; Meyerowitz, 1997). In contrast to their conserved functions, plant meristems develop highly diversified structures and morphology (Evert, 2006; Steeves & Sussex, 1989). In seed plants, shoot apical meristems and root apical meristems consist of clonally different cell layers and distinct functional zones (Gaillochet & Lohmann, 2015; Geng & Zhou, 2021; Han et al., 2020; Heidstra & Sabatini, 2014; Meyerowitz, 1997; Tsukaya, 2021). On the other hand, meristems in seed-free plant lineages usually maintain one single apical cell (AC), which is also called the apical initial or initial cell (Gifford, 1983; Harrison, 2017; Philipson, 1990; Rensing, 2017) and is unidentified in meristems of seed plants. ACs have unique tetrahedral or wedge shapes, and they have been actively studied for their roles in gametophytes of seed-free non-vascular plants (bryophytes) and in sporophytes of seed-free vascular plants (ferns and lycophytes) (Frangedakis et al., 2021; Harrison, 2017; Hata & Kyozuka, 2021; Philipson, 1990; Rensing, 2017). Specifically, gametophytes of bryophytes contain one or a few morphologically distinguishable ACs (Floyd & Bowman, 2007; Hata & Kyozuka, 2021). In the moss *Physcomitrella patens*, the persistent tetrahedral AC continuously sustains gametophyte development and leaf-like organ formation (de Keijzer et al., 2021; Harrison et al., 2009; Kofuji & Hasebe, 2014; Rensing et al., 2020; Véron et al., 2021). In the liverwort *Marchantia polymorpha*, the AC drives notch formation and promotes thallus expansion (Bowman et al., 2016; Hong & Roeder, 2017; Solly et al., 2017). In sporophytes of ferns and lycophytes, ACs maintain a similar function (Harrison, 2017; Plackett et al., 2015). For example, two adjacent apical initial cells promote cell proliferation and sustain shoot

development in the lycophyte *Selaginella kraussiana* (Harrison et al., 2007). Transcriptomes of the AC-type shoot apical meristems were previously characterized in the lycophyte *S. moellendorffii* and the monilophyte *Equisetum arvense* (Frank et al., 2015). In fern sporophytes, the single AC is sufficient to drive frond initiation and development in the fern *Nephrolepis exaltata* (Sanders et al., 2011), and the tetrahedral AC is also present in the fern *Ceratopteris richardii* (Hou & Hill, 2002). In fern gametophytes, however, the function and activity of ACs remain largely unclear. Gametophytes of different ferns develop different types of indeterminate meristems to sustain prothallus development and sexual reproduction, including the AC-based meristem and the multicellular meristem (Atkinson & Stokey, 1964; Imaichi, 2013; Nayar & Kaur, 1971; Raghavan, 1989). The AC-based meristem comprises the wedge-shaped AC and its immediate progenies located at the anterior part of the prothallus (Imaichi, 2013; Nayar & Kaur, 1971). In contrast, the multicellular meristems in fern gametophytes contain a group of adjacent narrow rectangular cells at the outermost layer, showing the cellular organization distinct from that in the AC-based meristems (Banks, 1999; Geng et al., 2022; Imaichi, 2013; Nayar & Kaur, 1971; Takahashi et al., 2009; 2012; 2015; Wu et al., 2021; 2022).

Previous studies demonstrated that the gametophytes of many examined ferns either lacked persistent ACs or maintained only an active AC for a limited time throughout the whole gametophyte development (Banks et al., 1993; Bartz & Gola, 2018; Conway & Di Stilio, 2020; Geng et al., 2022; Imaichi, 2013; Takahashi et al., 2009; 2012; 2015; Wu et al., 2021; 2022). In gametophytes of the widely studied fern *C. richardii* (Pteridaceae) (Banks, 1999; Bui et al., 2015; Chatterjee & Roux, 2000; Cooke et al., 1995; Eberle et al., 1995; Geng et al., 2021a; Geng et al., 2022; Hickok et al., 1995; Marchant et al., 2019; Plackett et al., 2015; 2018), the AC is only transiently present in a gametophyte after germination, and it quickly disappears as the prothallus expands (Banks, 1999; Bartz & Gola, 2018). Prothallus development in Ceratopteris is mainly driven by the multicellular meristem initiated from the lateral marginal layer (Banks, 1999; Bartz & Gola, 2018; Conway & Di Stilio, 2020), and this multicellular meristem does not contain any morphologically distinct ACs (Banks, 1999; Geng et al., 2022). Similarly, gametophytes of *Anemia phyllitidis* (Anemiaceae) also develop the multicellular meristem at the lateral side, lacking an AC (Takahashi et al., 2012). In gametophytes of three other fern species, *Lygodium japonicum* (Lygodiaceae), *Woodsia obtusa* (Woodsiaceae), and *Colysis decurrens* (Polypodiaceae), the ACs maintain themselves through the oblique division at early stages of gametophyte development (Imaichi, 2013; Takahashi et al., 2009; 2015; Wu et al., 2022). After only a few rounds of cell division, the AC-based meristem disappears and is replaced by the multicellular meristem in the same central apical region of the gametophytes (Imaichi, 2013; Takahashi et al., 2009; 2015; Wu et al., 2022). Interestingly, a recent study showed that some gametophytes of *Pteris vittata* (Pteridaceae) are able to maintain both the wedge-shaped AC and the multicellular meristem simultaneously at the different locations of prothalli. However, these ACs do not play a role in notch formation during prothallus development (Wu et al., 2021). All the previous findings suggest that ACs usually only contribute to the early development of gametophytes, and they disappear at late developmental stages (Imaichi, 2013). Considering highly diversified gametophyte morphology and developmental processes among fern taxa (Christenhusz & Byng, 2016; PPG I, 2016; Pryer et al., 2004; Sessa, 2018; Watkins et al., 2012), it remains unclear whether any persistent ACs exist in fern gametophytes to sustain

prothallus expansion and drive notch formation. Furthermore, the dynamic division and growth pattern of any persistent AC is also completely unknown.

We imaged and examined gametophyte development in different fern species at single-cell resolution to tackle these questions. Surprisingly, in a few previously uncharacterized ferns from the order Polypodiales, the wedge-shaped ACs can be identified even at the late stages of prothallus development, when the deep notch has been established with two wings fully expanded. Using *Sphenomeris chinensis* (Lindsaeaceae)—the lace fern (Hassler & Schmitt, 2019; PPG I, 2016)—as a representative species, we determined the growth dynamics of the AC and its immediate progenies during gametophyte development through time-lapse confocal imaging and quantitative analysis. In the gametophytes of *S. chinensis*, the AC, together with its immediate progenies, form a dynamically maintained cell packet. Quantitative results further demonstrated that active proliferation in the AC and its immediate progenies results in the increased number of small cells at the apical centre of gametophytes, contributing to prothallus expansion and notch formation. This study identified and quantitatively characterized the previously undefined ACs, which remain active at the late stages of fern gametophyte development, broadening our understanding of diversified meristem identities in land plants.

## 2. Results

### 2.1. Persistent ACs during gametophyte development in *Sphenomeris chinensis*

We took confocal microscopy snapshots and found that the wedge-shaped AC was continuously present in *S. chinensis* gametophytes at different developmental stages. At 29 days after inoculation (DAI) and 31 DAI, when the prothalli were composed of ~38 and ~55 cells, respectively, the wedge-shaped AC (indicated by red stars) at the centre of the apex was distinguishable from the other cells based on cell morphology (Figure 1a,b). At 37 DAI, *S. chinensis* gametophytes continued to develop as one single layer of cells. Interestingly, at 37 DAI, when the size of one *S. chinensis* prothallus increased to more than 180 cells with a deep notch at the apex, the wedge-shaped AC (indicated by the red star) was still present (Figure 1c). These observations suggested long-term maintenance of ACs in *S. chinensis* gametophytes. They further promoted us to determine growth and division patterns that drive AC proliferation.

### 2.2. Cell division patterns in the AC and its immediate progenies in gametophytes

To quantitively determine the division and growth patterns of the AC and its immediate progenies in *S. chinensis* gametophytes, we performed confocal time-lapse imaging and computational image analysis (Figures 2, 3, Supplementary Figures S1–S8, Supplementary Tables S1–S16), using the quantitative imaging platform that we have established (Wu et al., 2021; 2022). Each *S. chinensis* gametophyte was live-imaged at an initial time point (0 hr) and 48 hr after that (48 hr). These samples were visualized through the z-projection view of confocal stacks and were segmented using the established 2D watershed method (Vincent & Soille, 1991; Wu et al., 2022) to identify the cell outline (Supplementary Tables S1–S16). Then, the size of each cell in gametophytes was computationally determined and colour-coded (Figures 2b,d,g,i,3a2,a4,b2,b4,c2,c4,d2,d4,e2,e4,f2,f4). Cell divisions during the 48-h period were determined and mapped to

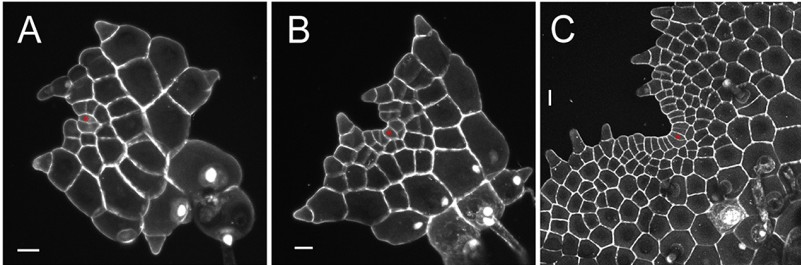

**Fig. 1.** Confocal microscopy snapshots showing the presence of apical cells (ACs) in *S. chinensis* gametophytes with the established apical notch. (a–c) Three *S. chinensis* gametophytes were stained and imaged through laser scanning confocal microscopy at 29 (a), 31 (b), and 37 (c) days after inoculation (DAI). (a–c) Grey: propidium iodide (PI) stain; scale bar: 20 μm. Red stars indicate the wedge-shaped apical cells. At least three independent biological replicates showed the ACs comparable to each representative snapshot in the figure.

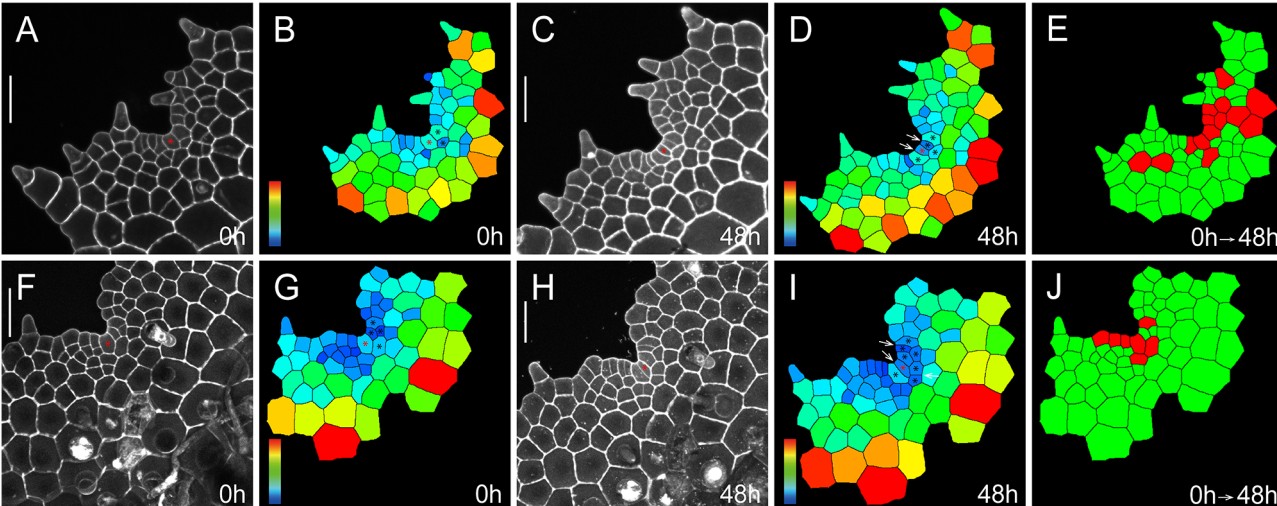

**Fig. 2.** Self-renewal of ACs in *S. chinensis* gametophytes. Two *S. chinensis* gametophytes (a–j) were stained and live imaged through laser scanning confocal microscopy at 0 hr (a,f) and 48 hr (c,h). (b,d,g,i) The computational segmentation and cell size quantification of confocal images in (a,c,f,h), respectively. (e,j) highlight cell division in the gametophytes (a,f), with the cells that divided during the analysed period (48 hr) shown in red and the cells that did not divide during the same period shown in green. (a) shows a gametophyte at 34 DAI, and (f) shows a gametophyte at 37 DAI. (b,g) highlight the conserved cell packet composed of an AC (indicated by a red star) and its immediate progenies (indicated by black stars) at 0 hr. (d,i) highlight the progenies of all the cells from the conserved cell packets at 48 hr, with the AC labelled with the red star. White arrows in (d,i) indicate the cell division in each packet. (a,c,f,h) Grey: propidium iodide (PI) stain; scale bar: 50 μm. Colour bars in (b,d,g,i) show the scales for the quantified area of each segmented cell, from blue (0) to red (≥ 700 μm²) in (b,d) and from blue (0) to red (≥ 1,200 μm²) in (g,i). At least three independent biological replicates showed the self-renewal of the ACs during gametophyte development.

the segmented images (Figures 2e,j, 3a5–f5). We found that the wedge-shaped AC, together with two adjacent trapezoid-shaped progenies, formed a unique three-celled packet in the wedge shape (Figures 2, 3, Supplementary Figure S9A). During prothallus development, this cell packet was dynamically maintained through a combination of cell division and expansion patterns. Specifically, the new wedge-shaped cell was produced through the conserved oblique division in an AC (Figures 2a–d,f–i, 3a1–a4, illustrated in Supplementary Figure S9A–E). The oblique division resulted in two daughter cells with a new wedge-shaped cell surrounded by a large trapezoid-shaped cell (Figures 2a–d,f–i,3a1–a5, Supplementary Figure S9B). Following that, a periclinal division occurred in the newly formed trapezoid-shaped cell, leading to a dynamic transition from the two-celled packet to a newly established three-celled packet at the centre of the developing notch (Figure 3e1–e5,f1–f5). These results suggest that dynamic maintenance of the AC in *S. chinensis* gametophytes is achieved through the combination of oblique and periclinal divisions in the individual cell packet. Unlike the wedge-shaped AC, the trapezoid-shaped cell within the three-celled packet underwent an anticlinal or periclinal division, giving rise to two new slender or short

trapezoid-shaped cells (Figures 2, 3a1–a5,b1–b5, Supplementary Figure S9A,D,E). In multiple gametophytes, the oblique division in the AC and the division (either in an anticlinal or periclinal orientation) in the adjacent trapezoid-shaped cell occurred within 48 hr, yielding five cells in the packet (Figures 2a–e, 3a1–a5, Supplementary Figure S9D,E). To reveal a spatial map of cell divisions during *S. chinensis* gametophyte development, we determined all the cells with or without cell division over 48 hr and visualized the pattern and location of these cells in the segmented images (Figures 2e,j, 3a5–f5). The division maps demonstrated that besides the conserved cell packet, cells surrounding the AC-containing cell packet also divided over the 48 hr (Figures 2e,j, 3a5–f5).

## 2.3. Size and position of the AC and its derivatives in gametophytes

We would like to know if the position of these cells determines the size of cells we analysed during the AC-driven gametophyte development. To reveal the potential correlation between the cell size and cell position, we first defined the centre of the AC-based

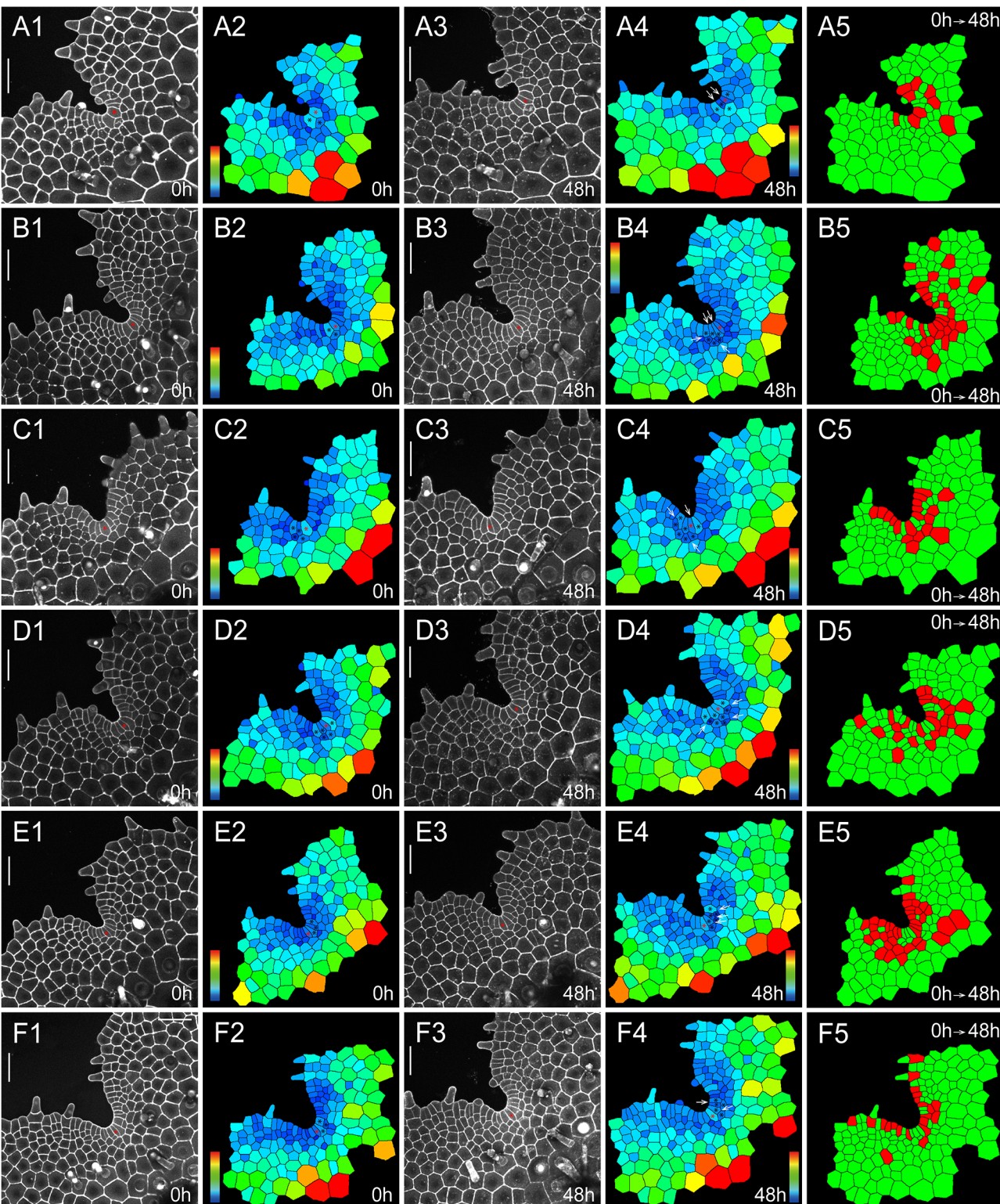

**Fig. 3.** The homeostasis of ACs and their immediate progenies during *S. chinensis* gametophyte development. Six *S. chinensis* gametophytes (a1–a5,b1–b5,c1–c5,d1–d5,e1–e5,f1–f5) were stained and live imaged through laser scanning confocal microscopy at 0 hr (a1,b1,c1,d1,e1,f1) and 48 hr (a3,b3,c3,d3,e3,f3). (a2,a4,b2,b4,c2,c4,d2,d4,e2,e4,f2,f4) The computational segmentation and cell size quantification of confocal images in (a1,a3,b1,b3,c1,c3,d1,d3,e1,e3,f1,f3), respectively. (a5,b5,c5,d5,e5,f5) highlight cell division in the gametophytes (a1,b1,c1,d1,e1,f1), with the cells that divided during the analysed period (48 hr) shown in red and the cells that did not divide during the same period shown in green. (a1,b1,c1,d1,e1,f1) show the gametophytes at 37 DAI. (a2,a4,b2,b4,c2,c4,d2,d4,e2,e4,f2,f4) show the conserved cell packet composed of an AC (indicated by a red star) and its immediate progenies (indicated by black stars). White arrows (a4,b4,c4,d4,e4, and f4) indicate cell division in each packet. (a1,a3,b1,b3,c1,c3,d1,d3,e1,e3,f1,f3) Grey: propidium iodide (PI) stain; scale bar: 50 μm. Colour bars in (a2,a4,b2,b4,c2,c4,d2,d4,e2,e4,f2,f4) show the scale for the quantified area of each segmented cell, from blue (0) to red (≥1,200 μm$^2$).

meristem in *S. chinensis* gametophytes, at which the wedge-shaped AC and its immediate progenies were located (circled in Supplementary Figure S10). We then considered these individual cells as the centre group (centre of the meristem) and compared them with all the other segmented cells (except trichomes) outside the centre group in gametophytes (Supplementary Figure S10). Our quantification in each independent sample (Supplementary Figure S10A–H) showed that at 0 hr, the average cell size in the centre group was significantly smaller than that of the cells outside the centre group (Figure 4a–h). In addition, at 48 hr, the average cell size of the centre group, which included all the progenies of the cells at 0 hr, was also significantly smaller than that of the segmented cells outside the centre group (Supplementary Figures S11A–H, S12A–H). The differences revealed by these statistical analyses can also be visualized from the colour-coded cell size maps, where small cells were coloured blue and large cells were coloured red (Figures 2b,d,g,i, 3a2,a4,b2,b4,c2,c4,d2,d4,e2,e4,f2,f4), showing the size gradient from small cells at the apical centre to large cells located distally to the apical notch, within the flat sheet of each prothallus.

## 2.4. AC and its immediate progenies serve as the active proliferation site

We wonder if the small cell size in the centre group is due to more active cell division, slower cell expansion, or both. Thus, we accessed the percentages of the cells divided from each live-imaged sample. Over the 48-hr period, the average percentage of cells in the centre group that divided (mean = 45.64%, *n* = 8) was significantly higher (*p* = .0002, two-tailed *t*-test) than that outside the centre group (mean = 15.10%, *n* = 8) (Figure 4i, Supplementary Table S17). This result can also be visualized from the cell division maps (Figures 2e,j, 3a5–f5), showing many cells at the centre of meristems divided over 48 hr (labelled in red). We then accessed the cell growth over the 48 hr in these samples by quantifying the changes in total cell area in the two groups (Figure 4j, Supplementary Table S18). Over the analysed time period, the relative total cell area (48/0 hr) in the centre group (mean = 157.63%, *n* = 8) was slightly higher than that outside the centre group (mean = 132.39%, *n* = 8), but the difference was statistically significant (*p* = .0011, two-tailed *t*-test) (Figure 4j, Supplementary Table S18). In addition, we also generated the colour-coded cell growth maps to visualize the relative growth of each segmented cell from the eight gametophytes over 48 hr (Supplementary Figure S13A–H). All these maps showed a gradient from the fast cell growth area (indicated by red, yellow, and green) surrounding the apical notch to the slow growth region (indicated by blue) that is located distally to the notch (Supplementary Figure S13A–H). Taken together, all the results from cell division and growth quantification suggested that the AC and its immediate progenies maintain active cell proliferation, leading to the increased number of cells with small size at the apical centre of gametophytes and eventually driving prothallus development.

## 2.5. AC maintenance in gametophytes of different ferns

We wonder if the maintenance of ACs in gametophytes is a unique character in *S. chinensis* (Lindsaeaceae) or a shared trait among different fern species, so we took confocal microscopy snapshots and performed cell size quantification of the gametophytes of representative ferns from different taxa (PPG I, 2016). Among the ferns we characterized, we found that the wedge-shaped

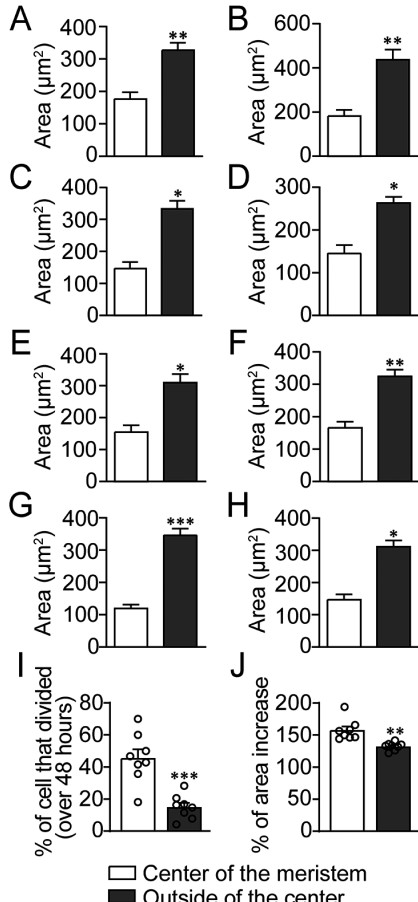

**Fig. 4.** Quantified cell size variation, division activity, and cell expansion in the meristems and gametophytes. (a–h) The area quantification of the labelled cells from each gametophyte (Supplementary Figure S10A–H), respectively, at 0 hr. (a–h) The Y-axis showed the average cell area. (a–j) The X-axis represented the centre of the meristem (white) and outside the centre (black). (a) *n* = 10 cells at the centre of the meristem and 49 cells outside the centre. (b) *n* = 14 cells at the centre of the meristem and 47 cells outside the centre. (c) *n* = 11 cells at the centre of the meristem and 89 cells outside the centre. (d) *n* = 10 cells at the centre of the meristem and 131 cells outside the centre. (e) *n* = 14 cells at the centre of the meristem and 88 cells outside the centre. (f) *n* = 15 cells at the centre of the meristem and 107 cells outside the centre. (g) *n* = 12 cells at the centre of the meristem and 112 cells outside the centre. (h) *n* = 12 cells at the centre of the meristem and 138 cells outside the centre. (i) Y-axis: the average percentage of the cells that divided over 48 hr from the centre of the meristem or outside the centre (*n* = eight independent samples shown in Supplementary Figures S10, S11). The cells at the centre of the meristems or outside the centre of each gametophyte were determined (Supplementary Figures S10, S11). (j) Y-axis: the average relative area (48/0 hr) at the centre of the meristem or outside the centre (*n* = eight independent samples shown in Supplementary Figures S10, S11). During the analysed period (48 hr), the relative total cell area (48/0 hr) from the centre of meristems in each gametophyte is calculated as the area of total cells (including daughter cells) at the centre of the meristem (at 48 hr)/the area of total cells at the centre of the meristem (at 0 hr). The relative total cell area (48/0 hr) from the cell group outside the centre is calculated as the area of total cells (including daughter cells) outside the centre (at 48 hr)/the area of total cells outside the centre (0 hr). Bars: means ± SEs. ∗∗*p* < .01, ∗ ∗ ∗*p* < .001 (Student's two-tailed *t*-test). The source data for Figure 4i,j are included in Supplementary Tables S17 and S18.

ACs were also present in the gametophytes of *Blechnum australe* (Blechnaceae) and *Cyrtomium macrophyllum* (Dryopteridaceae) at different developmental stages (Figure 5a–l), even when a prothallus contained more than 140 cells and developed a deep apical notch (Figure 5g–j). These results suggested that cell proliferation driven by the persistent AC, which was undercharacterized previously, likely serves as a mechanism underlying

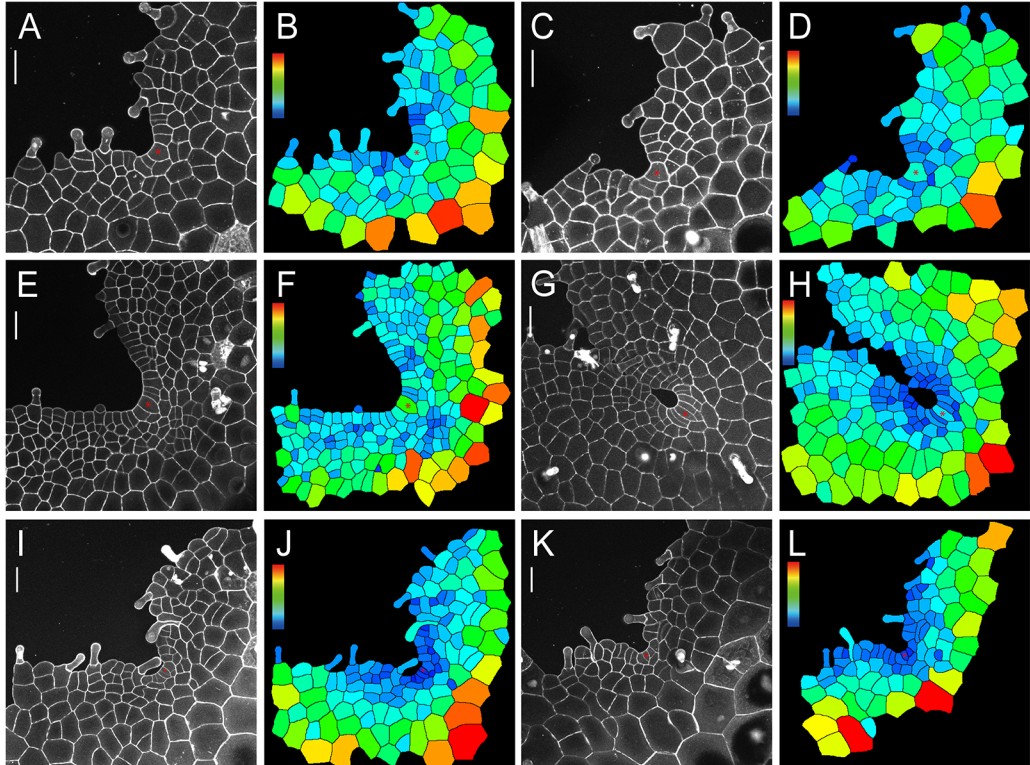

**Fig. 5.** The confocal imaging and computational analysis show the maintenance of ACs in gametophytes of *B. australe* and *C. macrophyllum*. Four *B. australe* gametophytes were stained and imaged through laser scanning confocal microscopy at 20 (a,c), 23 (e), and 31 (g) DAI. Two *C. macrophyllum* gametophytes were stained and imaged through laser scanning confocal microscopy at 36 (i,k) DAI. (b,d,f,h,j,l) The computational segmentation and cell size quantification of confocal images in (a,c,e,g,i,k), respectively. (a,c,e,g,i,k) Grey: propidium iodide (PI) stain; scale bar: 50 μm. Red stars indicate the wedge-shaped ACs. Colour bars in (b,d,f,h,j,l) show the scales for the quantified area of each segmented cell, from blue (0) to red (≥1,800 μm²) in (b,d,f), from blue (0) to red (≥2,500 μm²) in (h), and from blue (0) to red (≥3,200 μm²) in (j,l). At least three independent biological replicates showed the maintenance of an AC comparable to each representative snapshot included in the figure.

gametophyte development across different families in Polypodiales (Supplementary Figure S14).

## 3. Discussion

This study shows that the wedge-shaped AC is present and maintained in the gametophytes from three fern species in the order Polypodiales, even at the late developmental stages when the gametophytes have fully expanded and formed the deep notch at the apical centre (Figures 1b,c, 5a–h). Specifically, *S. chinensis* (Lindsaeaceae), *B. austral* (Blechnaceae), and *C. macrophyllum* (Dryopteridaceae) all showed similar morphology of ACs even after the establishment of the apical notch (Figures 1, 5, Supplementary Figure S14). These ferns belong to different suborders: Lindsaeineae, Aspleniineae, and Polypodiineae, respectively (De Gasper et al., 2016; Hassler & Schmitt, 2019; PPG I, 2016). On the contrary, we recently characterized *W. obtusa* (Woodsiaceae), a fern also belonging to the suborder Aspleniineae, and we found that *W. obtusa* gametophytes do not maintain any morphologically distinct AC at late developmental stages (Wu et al., 2022). These results altogether promote us to interpret that this trait likely independently evolved or had diversified among different fern taxa. Furthermore, these ACs are not mitotically quiescent, and they actively divide in *S. chinensis* gametophytes (Figures 2–4, Supplementary Figure S12). These results provided evidence supporting a long-standing assumption that the AC in fern gametophytes can continuously drive prothallus expansion and shape gametophyte

morphology (Nayar & Kaur, 1971), independent of multicellular apical or marginal meristems.

Through quantitative time-lapse confocal imaging, we determined cellular dynamics underlying the AC-driven proliferation in *S. chinensis* gametophytes, which is unique and distinct from the previously characterized cell growth patterns in gametophytes of *C. richardii* (Pteridaceae), *P. vittata* (Pteridaceae), *W. obtusa* (Woodsiaceae), *L. japonicum* (Lygodiaceae), *A. phyllitidis* (Anemiaceae) or *C. decurrens* (Polypodiaceae) (Takahashi et al., 2009; 2012; 2015; Wu et al., 2021; 2022). We demonstrated that at the apical centre of *S. chinensis* gametophytes, the AC and its adjacent immediate progenies form a conserved wedge-shaped cell packet (Figures 2, 3), which was dynamically maintained through the oblique division in the AC and the sequential periclinal and anticlinal divisions in its adjacent progenies (Figure 3b1–f5, Supplementary Figure S9). It has been well-documented that the oblique division plays a conserved role in renewing the ACs in gametophytes of *W. obtusa* (Woodsiaceae) (Wu et al., 2022), *L. japonicum* (Lygodiaceae) (Takahashi et al., 2015), *P. vittata* (Pteridaceae) (Wu et al., 2021), and *C. decurrens* (Polypodiaceae) (Takahashi et al., 2009). However, in the species examined previously (Takahashi et al., 2015; Wu et al., 2022), the oblique division was maintained for only a limited time, leading to the disappearance of wedge-shaped ACs and the replacement of the AC-based meristem by the multicellular apical meristem, which is different from the ACs in gametophytes of *S. chinensis* (Lindsaeaceae). Future studies focusing on the regulatory signals that sustain the oblique division in ACs will provide more insights

into AC maintenance in ferns. In addition, compared to meristems in angiosperm sporophytes, such as shoot apical meristems in Arabidopsis (Jones et al., 2017; Laufs et al., 1998; Louveaux et al., 2016; Meyerowitz, 1997; Shapiro et al., 2015; Willis et al., 2016; Yang et al., 2017), the AC-based meristems in fern gametophytes show distinct division patterns and activities. It will be interesting to determine whether shared or different mechanisms regulate cell divisions in these two different meristem identities.

Our quantitative data revealed that at the centre of the meristem in *S. chinensis* gametophytes, the cell size from the group—including the AC, its immediate progenies, and their adjacent cells—was significantly smaller than that of the cells outside the group (Figure 4a–h, Supplementary Figures S10–S12). In addition, the cell division activity and cell growth within this group are significantly higher than outside the group (Figure 4i,j, Supplementary Figures S10, S11). In addition, the colour-coded growth maps revealed that the region of active cell expansion is surrounding the apical notch, in contrast to the slow growth region located distally to the notch (Supplementary Figure S13). All these results uncover the relationship between the cell position, size, growth, and division activity in gametophytes, suggesting cell position plays a role during AC-driven cell proliferation in fern gametophytes.

## 4. Methods

### 4.1. Plant materials and growth condition

The spores of *S. chinensis* (order number: 1884), *B. australe* (order number: 440), and *C. macrophyllum* (order number: 744) were requested from and kindly provided by the American Fern Society (AFS). The spores of *S. chinensis* were propagated and generously shared by Dr. McNickle at Purdue. The spores were germinated on the pots containing SunGro horticulture propagation mix in continuous light at 25 °C. All the growth pots were covered and sealed with transparent plastic wrap to retain moisture.

### 4.2. Confocal live imaging and image analysis

To reveal the morphology of *S. chinensis*, *B. australe*, and *C. macrophyllum* at single-cell resolution, the snapshots of different gametophytes were taken using a Zeiss LSM 880 upright confocal microscope. Specifically, each gametophyte was transferred on the FM plates, stained with propidium iodide (PI), and rinsed with sterilized water two or three times before confocal imaging, as described in Wu et al. (2021; 2022). The confocal snapshots included: *S. chinensis* gametophytes from 29 days after inoculation (DAI) to 37 DAI, *B. australe* gametophytes from 20 DAI to 31 DAI, and *C. macrophyllum* gametophytes at 36 DAI. The confocal imaging setting was described previously in detail (Geng & Zhou, 2019; Wu et al., 2021; 2022; Zhou et al., 2018).

Time-lapse confocal imaging was performed as described previously (Wu et al., 2021; 2022) to quantitatively determine the growth and division patterns of the AC and its immediate progenies during *S. chinensis* gametophyte development. A 48-hr period was used in the time-lapse imaging based on the cell number and growth and gametophyte morphology indicated by the confocal snapshots (Figure 1a–c). In total, 19 independent samples were live imaged at 0 and 48 hr, all showing active proliferation of ACs and their immediate progenies. Among them, eight samples were segmented and quantitatively analysed, shown in Figures 2a–j, 3a1–a5,b1–b5,c1–c5,d1–d5,e1–e5, and f1–f5. All the samples (imaged at 37 DAI or 34 DAI) showed comparable cell division

and growth patterns. The Image J / Fiji was used for generating the maximum intensity Z projection view of confocal images (Schindelin et al., 2012). The 2D image segmentation of the confocal images was performed following the established method described by Vincent and Soille (1991) and Wu et al. (2021). As shown in Supplementary Tables S1–S16, the area of each segmented cell from the gametophytes imaged at 0 and 48 hr was quantified using MATLAB software (2020b). The scale of each colour bar in each figure was specified in figure legends. The cells that divided or did not divide during the analysed period were determined based on the segmented time-lapse images over 48 hr and were labelled using MATLAB.

### 4.3. Quantification and statistical analysis

The centre of the meristem in each gametophyte was defined and included in the quantification (Figure 4, Supplementary Figures S10–S12). At 0 hr, the centre of the meristem—the centre group (highlighted with purple circles in Supplementary Figure S10)—included the wedge-shaped AC (labelled by red stars) and its adjacent progenies (labelled by black stars) and their surrounding cells (labelled with white dots). At 48 hr, the centre of the meristem (highlighted with purple circles in Supplementary Figure S11) included all the cells and their progenies from the centre group defined at 0 hr. All the other segmented cells (except trichomes) from each gametophyte were included in quantification as the group outside the centre (Figure 4, Supplementary Figures S10–S12). The division activity of the cells (except trichomes) at the centre of meristems or outside the centre in gametophytes (Figure 4i) was quantified using the following equations:

The percentage of cells that divided over 48 hr (at the centre of the meristem) = Number of cells (at the centre of the meristem) that divided over 48 hr / Number of total cells (at the centre of the meristem) at 0 hr x 100%;

The percentage of cells that divided over 48 hr (outside the centre) = Number of cells (outside the centre) that divided over 48 hr / Number of total cells (outside the centre) at 0 hr x 100%.

The increase of cell area in the meristems and gametophytes (except trichomes) (Figure 4j) was quantified using the following equations:

The relative total cell area (48/0 hr) in the centre of the meristem = area of total cells (including daughter cells) from the centre of the meristem (at 48 hr) / area of total cells from the centre of the meristem (at 0 hr) x 100%;

The relative total cell area (48/0 hr) outside the centre group = area of total cells (including daughter cells) from the group outside the centre (at 48 hr) / area of total cells from the group outside the centre (at 0 hr) x 100%.

To generate relative cell growth maps (Supplementary Figure S13), the area of each segmented cell at 0 and 48 hr (shown in Supplementary Tables S1–S16) was first calculated. Next, based on manually curated cell lineage information, the relative growth of cells was calculated using the cell areas data at 0 and 48 hr accordingly, using the following equation: The relative growth of each cell = [the cell area (including the area of its progenies) at 48 hr – the cell area at 0 hr] / the cell area at 0 hr. For the cell divided during the time interval, areas of its daughter cells were added up as the final cell area. Finally, the relative cell growth was plotted onto the segmented cells at 0 hr using MATLAB. The relative growth of each cell was quantitatively indicated by colours, with a scale from blue (0) to red (≥1). Blue indicates zero relative growth, and red indicates 100% or above relative growth during the 48-hr interval.

## 4.4. Construction of the phylogenetic tree

The phylogenetic tree of nine fern species with the characterized AC division and growth in their gametophytes (Banks et al., 1993; Geng et al., 2022; Takahashi et al., 2009; 2012; 2015; Wu et al., 2021; 2022) was generated, through PHYLOT (https://phylot.biobyte.de/index.cgi) as described previously (Geng et al., 2021b). Based on the search results from the NCBI taxonomy database (https://www.ncbi.nlm.nih.gov/Taxonomy/Browser/wwwtax.cgi), the homotypic synonym *Odontosoria chinensis* for *S. chinensis* and the homotypic synonym *Leptochilus decurrens* for *C. decurrens* were used as NCBI taxonomy IDs, for constructing the phylogenetic tree (Supplementary Figure S14) through PHYLOT.

## Acknowledgements

The authors would like to thank the American Fern Society, Brian Aikins, and Gordon McNickle for kindly sharing the spores used in this study. The authors also thank Purdue Bindley Bioscience Facility for the access to the ZEISS LSM880 confocal microscope.

**Financial support.** This work was supported by funds from Purdue Centre for Plant Biology and by the National Science Foundation (grant number IOS-1931114) to Y.Z.

**Conflicts of interest.** The authors declare no conflict of interest.

**Authorship contributions.** Y.Z. conceived the research direction, X.W. performed all the live imaging experiments, X.W., X.L., S.Z., and Y.Z. discussed the experimental results, A.Y. and Y.Z. performed image analysis, X.W. performed the quantification and statistical analyses, X.W. and Y.Z. wrote the manuscript, A.Y., X.L. and S.Z. revised the manuscript, and all authors approved the manuscript.

**Data availability statement.** The data that support the results and conclusions of this study are available within the paper and its supplementary materials. Any other supporting information and the code are available from the corresponding author upon request.

**Supplementary Materials.** To view supplementary material for this article, please visit http://doi.org/10.1017/qpb.2022.21.

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
