## [Reviewer Report]

Dear Editor,

We are submitting the manuscript titled “Quantitative live-imaging reveals dynamics of apical cells during fern gametophyte development” for publication in Quantitative Plant Biology as a research article.

All multicellular organisms face the same problem: how to specify and maintain different cell fates during continuous growth and development. Plant meristems represent an ideal model to address this fundamental question, because they contain self-renewing stem cells that maintain themselves undifferentiated, while continuously adding new cells that eventually form differentiated organs. 

Land plants alternate between the generations of asexual sporophytes and sexual gametophytes. Sporophytes of seed plants develop meristems that sustain organ formation and shape plant architecture, while gametophytes of seed plants are dependent on their sporophytes and are lacking meristems. In contrast, gametophytes and sporophytes of seed-free vascular plants, including ferns, are independent of each other, and fern gametophytes develop pluripotent meristems to drive independent growth. Furthermore, meristems of seed-free plants usually develop one single tetrahedral (pyramid-shaped) or wedge-shaped apical cell (AC) as the initial, which is lacking in seed plants. To date, it remains unclear how the AC promotes cell proliferation in fern gametophytes. It is also unknown whether any long-lived AC exists in fern gametophytes to continuously sustain prothallus development and determine gametophyte morphology. 

To tackle this question, we used Sphenomeris chinensis, the lace fern as a research system to quantitatively determine the AC dynamics in fern gametophytes. Frist, we identified the long-lived AC during Sphenomeris chinensis gametophyte development, which is different from the previously characterized, transiently present ACs in gametophytes of many other fern species. Through confocal time-lapse imaging, computational image analysis, and quantification of cell size, expansion and division activity, we then determined unique patterns of cell growth and division that maintain the homeostasis of the persistent AC and its immediate progenies. We also uncovered a linkage among the small cell size, active cell division and expansion, and cell position in fern gametophytes, suggesting a previously undefined mechanism underlying AC proliferation in fern gametophytes. Furthermore, we found the maintenance of long-lived ACs in gametophytes, though not conserved for all the fern species, is a shared character among different fern taxa. 

All these findings provide quantitative insight into the cellular basis and evolution of meristems in seed-free plant lineages, and they suggest both conserved and diversified mechanisms underlying meristem and gametophyte development across land plants. We think this work is of broad interest to the general plant biology research community.

Thank you so much for your consideration!

Best,

Yun Zhou Ph. D.

Assistant Professor

Department of Botany and Plant Pathology

Center for Plant Biology, Purdue University

---

## [Reviewer Report]

*Comments to Author*: Plant tissues and organs are generated from meristems, in which stem cells continuously divide to self-renewal, and differentiate to form various cell types. While the structure and molecular control of meristem development have been well studied in seed plants, the functions and cellular dynamics of meristems in early land plant lineages have remained unclear. Instead of harbouring a multicellular meristem containing cells with distinct identities, the apical cells (ACs) have been identified in bryophytes and ferns, which are considered to be the initial cells contributing to organ growth. This study investigates the dynamics of ACs during gametophyte development in Sphenomeris chinensis. The authors took advantage of time-lapse live imaging to track cell growth and division at the apical region of S. chinensis prothallus. By segmenting cells at two different time points and quantitative analysis, they showed that a small group of cells surrounding the ACs exhibited high proliferation activity and smaller cell size. Based on the observation that ACs could be detected in prothallus that contain many cells, they propose that ACs are long-lived and could be maintained until late developmental stages.

The manuscript is well written and sound. The findings provide some insights into the dynamical behaviours of fern meristems, thus may help to advance our understanding of meristem functions in the context of evolution. The methodology of time-lapse imaging may pave a way for genetic and functional analysis using fern as a model at the cellular level.

Following are my comments and suggestions for the authors to consider:

1) The ACs are distinguished based on its tetrahedral or wedge shapes. But I found it not always easy to tell, especially at late stages. The authors may need to give a more detailed description regarding how they define the ACs. And what could be helpful is to clearly label these cells as some are missing in the current figures.

2) The authors proposed that “active cell division instead of reduced cell expansion” led to smaller size of the ACs and its surrounding cells. Cell division could be readily seen from the data shown in Figs 2 and 3. However, a cell growth map would be required to support the conclusion of “reduced cell expansion”. This should be feasible given all the cell size data are available.

3) The length of plant cell cycle, although variable between cell types, is shorter than 48 hours. Why was this time interval was used? It seems a bit longer compared to normal plant cell division cycle and needs to be clarified.

4) Due to the lack of characterized molecular markers, the authors defined a group of cells as “the center of the meristem”, which includes ACs and their progenies. But the number of cell layers within this region (as labelled with circles) are not always consistent between different plants.

5) The stem cell initials, such as the WUS-expressing cells in Arabidopsis shoot apical meristem, usually maintain a very low mitotic activity whereas their progenies at the central zone and peripheral zone divides more actively. If the AC in fern gametophytes was able to continuously divide thus to drive prothallus expansion, how its own identity could be maintained needs to be discussed.

6) The gametophytes shown in the data were from various growth stages (e.g., 34 DAI, 37 DAI). I am not sure whether this needs to be consistent throughout the manuscript. While emerging as a new model, the morphologies of fern during development are still not familiar to many researchers particularly those outside of the field. Therefore, in addition to the confocal images, photos of S. chinensis prothallus at different growth stages might be useful for the readers to gain a better idea what the segmented cells represent for.

7) In the caption Figure 2, “Red circles in (B, D, G, I) indicate the conserved cell packet composed of an AC and its immediate progenies”. There are no red circles on the images.

8) Some images appear to be redundant with others, which can be combined or removed to the supplementary data in order to make the figures more concise.

---

## [Reviewer Report]

*Comments to Author*: Wu et al. propose a quantitative description of Sphenomeris chinensis gametophyte morphology and growth using confocal imaging and computational analysis methods. Overall, the article is simple and well written, and provides new information about meristem function and apical cell maintenance in ferns.

Minor points :

• Figure 3 legend : “red circles” should be replaced with “asterisks”. In panel (G), some new cell walls are not shown in red.

• Lines 142-179 : It would be nice to indicate somewhere which cell divisions are described as it is hard to connect the text and the figures. For example, authors could number their little drawings in panel G (i, ii, iii, iv…) and refer to them directly in the main text.

• Data provided in Fig S10 is interesting but has not been quantified as in other figures. This should be done and the new data could be presented in a sixth main figure.

• A clear and objective definition of a “long-lived apical cell” is lacking. How many cells should have the gametophyte and how old should it be to be considered as having a “long-lived apical cell”?

• The origin and quality of the biological material is unclear. Has the material been deposited in a herbarium?

---

## [Reviewer Report]

*Comments to Author*: Dear authors, 

We thank you for your appreciated manuscript presenting a meaningful dynamic description of apical cells in fern gametophyte. Your manuscript has been now revised by two reviewers (please find their comments below), with interesting comments which would improve the manuscript. Would you please prepare a corrected version and point by point response taking into account the modifications suggested by the reviewers ? 

Thank you very much in advance, 

Looking forward to reading you

Best regards

Daphné Autran

---

## [Reviewer Report]

Dear Editor,

Thank you so much for sending us the reviewers’ comments, which help us improve the manuscript. We have carefully revised the manuscript following all the comments and suggestions from the editors and reviewers. 

We are submitting the revised manuscript titled “Quantitative live-imaging reveals the dynamics of apical cells during gametophyte development in ferns” for publication in Quantitative Plant Biology. 

In this submission, please find:

• the response letter addressing all the reviewers’ comments point by point, 

• Graphic abstract,

• the revised main text with the revised title,

• Figures 1-5, 

• the supplementary file containing Supplementary Figures S1-S14,

• Supplementary Tables S1-S18.

Thanks again for taking the time to consider our revised manuscript!

Best,

Yun

Yun Zhou Ph. D.

Associate Professor 

Department of Botany and Plant Pathology

Center for Plant Biology, Purdue University

Email: zhouyun@purdue.edu

---

## [Reviewer Report]

*Comments to Author*: In the current version, the authors have analysed the size of the cells in fern gametophytes and generated a growth map according to my suggestions. They have also extended the description of the methods, which is now clear and precise. I think my concerns are adequately addressed and the manuscript is now suitable for publication in Quantitative Plant Biology.

---

## [Reviewer Report]

*Comments to Author*: Dear Dr Zhou and colleagues, 

Thank you for all the work in revising your manuscript. We have received the reviewers’ comments on the new version, acknowledging the careful revisions and new data added following their suggestions. I fully agree with their recommendation to accept your manuscript for publication in Quantitative Plant Biology. This work will certainly enhance the interest on the morphodynamics of apical stem cells and pave the way for quantitative imaging studies in non-model plant species. 

Best regards, 

Daphné Autran